global mental health; gender; community-based initiatives; decolonization; rural health

**Corresponding author:**
Ana Cecilia Ortega;
Email: ortega.anaceci@gmail.com

# To gather is to heal: Women's mental health circles in rural Chiapas, Mexico

Ana Cecilia Ortega[1] and Margaret Buckner[2]

[1]Compañeros en Salud, Chiapas, Mexico and [2]Laboratoire d'ethnologie et de sociologie comparative (Université de Paris – Nanterre), Compañeros en Salud, Missouri State University, Maryville, MO, USA

## Abstract

In the rural villages of the Sierra Madre region of Chiapas, women experiencing hardship show signs of emotional distress that are diagnosed as depression and anxiety by health professionals. In this study, we critically analyze the impact of a pilot mental health group intervention (Women's Circles) facilitated by community mental health workers. The intervention consisted of eight structured sessions that included psychoeducation from a gender perspective, mindfulness exercises, interactive activities, arts and crafts, and sharing personal experiences. We carried out participant observation and 27 semi-structured interviews with the participants. The main outcomes were, first, that participants' moods improved, and second, that the improvement was mainly due to gathering with others and having someone to talk to. In addition, we observed that lessons during the Circles were often prescriptive, which, rather than creating a space for reflection on personal experiences, imposed globalized views of mental health and gender. In sum, we describe both the positive impact this program had on mental well-being and the problematic spreading of psychoeducation.

## Impact statement

In this article, we contribute to the critical literature on global mental health by analyzing a pilot mental health group intervention in rural Mexico, called 'Women's Circles,' facilitated by community mental health workers (CMHWs). Through a psychology–anthropology collaboration, we question the universal benefit of psychoeducation. With a global call for organizations to increase access to mental health services and professionalize local CMHWs, this study serves as evidence to reflect critically on the training that CMHWs receive as well as the dominant mental health discourses and interventions they use. Our findings suggest that, when critically implemented, the Women's Circles, led by local CMHWs, might have the potential to serve as a nonpathologizing psychosocial well-being model in other underserved areas.

## Introduction

Since 2012, Compañeros en Salud (CES), the Mexican affiliate of Partners in Health (PIH, a global health non-profit organization based in Boston, MA) has been supporting rural health clinics in communities of 1,000–2,000 inhabitants in the Sierra Madre region of Chiapas, which, according to the CONEVAL (2020) is one of the most impoverished states in Mexico. This remote mountainous area suffers from scanty and poor roads, precarious income from coffee farming (due to rust, drought, soil depletion, and fluctuating prices), poor school quality, few employment opportunities, and an overall lack of government services, including health services. CES staffs ten rural Ministry of Health clinics with physicians in their final year of medical school (*pasantes*) and nurses and works with around 100 female community health workers, all of whom CES trains and supervises in clinical delivery and global health. They also provide medicine, transportation, and hospital referrals for advanced treatment.

In 2014, CES began a program to address the dearth of mental health services in Mexico, especially in rural areas (Berenzon Gorn et al., 2013). In this program, physicians are trained and supervised in the delivery of mental health care in the clinics and refer people diagnosed with depression and anxiety to community mental health workers called '*Cuidadoras de Salud Mental*' (mental health caregivers) (Rodríguez Cuevas et al., 2021). The Cuidadoras are trained to deliver Problem Management Plus (PM+), an individual, five-session, structured, psychological intervention that includes psychoeducation, screening for depression and anxiety, relaxation and problem-solving exercises, engaging in enjoyable activities, and strengthening social support networks (WHO, 2018). The Cuidadoras also receive training in other mental health topics, such as gender-based violence, trauma, addiction, grief, and psychological first aid.

Recent studies in some of the villages where CES works have reported that depression and anxiety are higher among women than men and are more prevalent than the national average (Elliot et al., 2019; Serván-Mori et al, 2021). It is also more common locally for women to seek mental health services than for men to do so. Perceived symptoms of depression and/or anxiety can be brought on by worries about domestic abuse and/or a controlling spouse (Aguerrebere et al., 2021), being unable to satisfy their family's financial and emotional needs (Hartman et al., 2023) and feeling isolated and lonely in their homes (Deitz et al., 2020). In this sense, this embodiment of social stressors is a form of what Kleinman et al. (1997) call 'social suffering.'

Therefore, to provide spaces for women to care for their mental health collectively, while reflecting on the underlying factors that affect their emotional well-being, CES designed and implemented 'Women's Circles' in four of the ten rural villages it supports. This intervention was inspired by the women's circles in Guatemala facilitated and codesigned by community health workers (Chomat et al., 2019) and was intended to improve participants' mood (ánimo), strengthen social networks, and reflect on mental health issues from a gender perspective.

The Women's Circles pilot program began in 2022. The Cuidadoras in the four villages participated in the program design and facilitated the Circles. The Circles were a structured group intervention, guided by a manual, with eight two-hour, bi-weekly sessions. Each session comprised psychoeducation talks on topics such as self-esteem, self-care, self-compassion, assertiveness, grief and loss, social support, alcohol abuse, gender roles, and violence against women; interactive and role-playing activities; questions to promote sharing of personal experiences; relaxation and mindfulness exercises; arts and crafts; and refreshments. At the end of these eight sessions, the participants received a diploma and had a graduation ceremony and celebration.

Most of the women invited to attend the Circles had been diagnosed at the local health clinic with depression and anxiety and had undergone PM+ sessions with the Cuidadoras. However, in some cases, the Cuidadoras and/or participants themselves invited other women who were not mental health patients they thought would be interested. Most of the women were married, some were widows, and the majority had children. Most were not formally employed but were full-time homemakers, though some also ran a small business from their home such as selling groceries or prepared food.

In this article, we used an ethnographic approach to critically examine the experience of the women who participated in these groups. As Jain and Orr (2016) highlighted, nuanced ethnographies can contribute to evaluating the achievements of global mental health projects, and that 'close observation and engagement with the field can reveal the dynamics through which mental health policy agendas play out on the ground' (p. 689).

Though we began this evaluation with no particular theoretical perspective in mind, upon analysis and reflection, we find that our results align with myriad authors (among others: Szasz 1960, 1994, 2007; Watters, 2010; Fernando 2014; Jenkins 2015; Duncan 2017) who question the validity of global mental health remedies and indeed the very fact of universal 'mental illness.' Moreover, the collaboration between a psychologist and an anthropologist (as advocated by Jenkins [2018]) proved invigorating, as each was challenged to reexamine her own biases and gained a new understanding of the other's discipline. Both not only experienced the sheer weight of the global authority of meta-psychoeducation and the assumed universal diagnoses of 'mental illness,' but also

witnessed the CES mental health team's intense dedication to improving the lives of women using the psychological methods they had been taught and their efforts to adapt them to the local context.

## Methods

Once the first Circles were completed, we created – with the Cuidadoras' participation – a semi-structured interview guide consisting of 13 questions covering a variety of themes: overall experience, perception of Cuidadoras' facilitation, what they learned, whether they shared or applied these lessons to their lives, whether they had made friends, what they liked the most and least about the sessions, difficulties in attending, and village hearsay about the Circles. The interviews were carried out more as open-ended dialog than as a strictly question-and-answer process, which led to free-flowing narratives and unsolicited comments. The women were allowed – even encouraged – to go off on tangents and to speak as much or as little as they wanted on a given topic. Almost all the participants seemed to enjoy being interviewed and were eager to share their experience; only two seemed hesitant and gave short, succinct answers. The interviews ranged in length from 8 to 45 min, with an average of 16 min, above and beyond the time it took for introductions, to explain the project, and to obtain informed consent.

We aimed to conduct interviews with all 31 participants who had attended at least 3 sessions, were at least 18 years old, and had not been interviewed recently for other research projects. Of these women, we were able to interview 27 in private settings, usually in their home. The remaining four could not be interviewed because they were away from home or there were not two witnesses to participants' signing the written informed consent (as required by the Ethics Committee who approved the study). The interviews were carried out (in Spanish) 3 to 4 months after the Circles ended and were audio-recorded and transcribed. We used Dedoose qualitative management software loosely to store and organize the transcriptions and then inductively group excerpts by themes. The deductive nature of the interview guide was counterbalanced by the breadth of the women's responses and participant observation, which allowed for inductive analysis. Some of the themes were not the same as the ones pre-defined in the interview guide.

The two authors also carried out participant observation in the four villages, where they talked with women and community members informally and observed interactions. The first author also observed and participated in several Circles sessions. The second author conducted the interviews. Both authors stayed in community members' homes or the local health clinic three to 4 days at a time, though not at the same time; the second author did not wear the organization's t-shirt and made it clear that she was not part of the clinical team. The authors also interacted with the Cuidadoras in both formal and informal settings; in particular, they presented initial findings from the interviews to the Cuidadoras as a group for their comments and feedback.

Both authors recognize their respective positionality and acknowledge that findings should be read as situated and partial knowledge (Haraway, 1988). They are both white cis-gender women. The first author, a Mexican psychologist born in Mexico City, has worked as staff in the CES mental health program since 2019. She supervised the Cuidadoras, led the design and implementation of the women's circles, and created the study protocol.

She has ample experience working with gender-based violence survivors through a feminist psychology lens. The second author, a United States anthropologist, has volunteered off and on for CES since 2018 (for over 2 years in the field) in monitoring and evaluation and as a researcher for several programs.

The project was approved by the Ethics Committee of the Instituto Mexicano de Trasplantes (July 18, 2022, IMT-05072022-01).

## Findings

### The importance of gathering and sharing

First and foremost, all the women said they found the Circle helpful and were pleased with the Cuidadoras' facilitation. They referred to the Circles as a positive experience and said they hoped to continue participating and that more women should be invited. Most of the participants recalled that before the Circles they had symptoms such as feeling sad, tired, or irritable; they cried a lot; they had body aches and trouble sleeping; and/or they suffered from low self-esteem. In two cases, they had suicidal thoughts, which they said the Circle helped them overcome.

Our main finding, however, is that this positive impact did not occur for the expected reasons, namely psychoeducation and structured activities. What helped the women the most was simply gathering and spending time together (*convivir*), which gave them space and time to get to know each other, share experiences, escape from their daily routine as housewives, do arts and crafts, and have fun together.

Many women mentioned the importance of listening to others telling stories similar to their own because they learned that they were not alone. They said they felt inspired by what others had done to move forward. The women felt supported in the group, and many were uplifted by the words of encouragement from others. One woman said, 'I very much liked the Circles, listening to other women talking about their experiences. Truly, listening to other women's stories really helped me realize that I'm not the only one and that there are solutions.'[1] Another stated, 'Their stories, yes, we cheer each other up. And despite illnesses and problems, we really encourage each other, because it's important to leave it all behind and begin a new life ahead.'

The participants greatly valued the confidentiality agreement, thanks to which they developed enough trust in each other to share personal issues that they otherwise would not have. For example, one woman said 'Having trust in the group, being able to talk about personal situations, getting things off our chests, telling the good as well as the bad, sharing how you feel with the others, feeling relaxed, crying … we had never had this kind of opportunity to talk (*platicar*) … Sometimes we live in the same village, but we don't know why things are happening to other people.' Within the group, they were able to share opinions safely, which is hard to do in public settings where men are present: 'I'm glad we could open up, that we shared ideas and thoughts openly, freely, trusting the group. I liked that. In our experience, and as we were raised, it's like 'you don't say anything, here it's only the man,' but there [in the Circles] it's different.' The majority also mentioned feeling relieved by talking about their feelings and getting things off their chest (*desahogar*). One woman said, 'As I talked openly and unburdened myself, I felt the sadness lifting little by little, going away.'

Gathering for a couple hours every 2 weeks also gave women a chance to get away from the house and leave their worries behind. In the communities where these Circles were facilitated, women are responsible for housework (cooking, cleaning, laundry), childcare, husband care, and often elder care, while men leave the house daily to work in the coffee fields or other jobs. Women rarely have an opportunity to visit with women other than relatives and in-laws. Many women were grateful for the Circles because it gave them an excuse to leave the household drudgery behind, a relief from the stress of completing their chores. Some explicitly mentioned feeling lonely and longing for the support of others. For example, in the words of one woman, the Circles helped 'take my mind off things, housework, because when you're at home, sometimes you're never done, there's always something else to do. Being at the Circles, I get away from those things and can have fun.'

The Circles allowed the women to make friends and connect with women outside the family, which is hard to do in these villages where women staying home is the norm. One woman emphasized how lonely she felt before joining the Circles: 'I was doing very poorly emotionally. I felt really bad, I felt alone. You think you're alone with your problems. I was feeling really low. I hadn't slept in a month, either at night or during the day.' Another specifically mentioned needing friends: 'We have families, but having friends is different. Sometimes your father or your mother or your brother, you can't talk with them about what you're feeling. Why? Because you're afraid … how shall I say it? … that they might judge you or say, 'No, that's not good.' Right? On the other hand, a friend, an unconditional friend, the first thing they tell you is, well, it's your decision, it's all right.'

Last but certainly not least, the women said they had fun. When specifically asked the question, 'what did you like most about the Circles?', the most common answer by far was arts and crafts (materials paid for by CES). In fact, according to some Cuidadoras, the participants would not have come if it weren't for the arts and crafts. Most women also mentioned refreshments, which were mostly provided by CES and prepared in advance by the Cuidadoras.

### Learning 'about' mental health

As opposed to sharing and having fun, information presented on mental health – self-care, self-esteem, assertiveness, dealing with negative thoughts and grief – was rarely mentioned. One participant even stated, 'They teach us, but it doesn't stick,' which illustrates that much of what was 'taught' was foreign, did not apply to them, or was not engaging; this was especially true of the women who could not read, although the Cuidadoras tried to adapt the few written exercises verbally or with pictures. We noticed that when asked what they'd learned, the women, rather than their own reflections and what they heard from other women in the Circles, repeated back what they had been taught by the Cuidadoras: that men and women have the same worth and should be treated equally; women's and children's rights; how men and women with mental health problems react differently (men drink whereas women hold their sadness inside and cry); how men are *machista* (sexist); how husbands 'should' treat their wives; how the women also had a right to go out with friends. One participant recalls her learnings: 'We [women] should be able to go out for a while … There [in the Circle] I also learned about gender. It's not because you are a woman that you are not going to contribute to the household. It's a woman's right … Women have always been rejected, have been pushed aside. Depression and anxiety affect women more.'

---

[1] All of the direct quotes in this section are from the Circles' participants interviewed.

In fact, there seemed to be a disconnect between what the Cuidadoras taught and what the women lived. For example, in their talks, the Cuidadoras teach that men are *machista*, that they drink, that they mistreat their wives, and that they do not treat their wives as equal. However, most of the women said their husband is supportive and understanding, that he does not drink or mistreat her, that the two communicate well, and that they share in decision-making. One woman stated, 'They tell us that we should go out, that not just the man should go out, that we women should also have a say, not just the man. The man should not put us down. We should always make decisions equally or come to an agreement. (…) My husband has always supported me, has treated me … he helps me. But for some people that's not the case. For some people, "No, you can't go out."' Although in some Circles participants shared experiences of domestic violence, these situations seemed to be mostly in the past.

Moreover, women said generally that they first heard about 'depression' and 'anxiety' from clinic doctors, from Cuidadoras in one-on-one PM+ therapy sessions, or from Cuidadoras in the Circles. For example, 'The doctors told me it was stress, or anxiety, or depression, because all I did was cry, just cry, and I wanted to die, I didn't want to live any more… I felt listless (*apagada*), I felt that for me there was no happiness, for me there was no reason for laughter.' So, what the women learned about mental health and how it relates to gender came largely from lessons taught by the *Cuidadoras* (or doctors) rather than their own observations and experiences.

### Applying and sharing what they learned

Participants were also asked what things they learned in the Circles that they practiced at home and whether they shared what they learned with others. By far, the most common answer for what they took home was relaxation exercises. The women said they used these exercises to improve their sleep, ease anger and stress, relieve physical symptoms, and avoid negative thoughts. One woman said, 'I had a lot of pain throughout my body and was feeling desperate because I couldn't do my housework. When I began doing the exercises they taught us [in the Circles], to relax, to breathe, to calm down, and the rest, the pain started going away a little.' Another woman related, 'What I learned there is that when sometimes I felt an anxiety attack or I started to feel sick, what I did was to position my legs the way they taught us and start breathing deeply, and with that I felt that it calmed me.'

When asked whether they shared what they'd learned with others, most said they had, particularly with husbands, but also family members and neighbors. In fact, two participants said they'd told their husbands what they'd learned about gender equality, which, they said, resulted in more equitable gender roles at home. Others shared relaxation exercises or what they had learned about anxiety and depression. For example, one woman said, 'I started talking to [my neighbor], telling her it was up to her, that she could control herself, and I started telling her about the relaxation exercises we had learned.' Another participant said, 'A woman came by and I asked her what was wrong. She said, 'I don't know why, but I feel weak, like there's something wrong inside my body (*fiereza*).' I told her that it was anxiety and depression. And since her father had just died, I told her it was anxiety.' Finally, two participants told friends experiencing intimate partner violence to defend themselves. One said that one of her friends is sometimes beaten by her husband because he comes home drunk. She told her friend, 'Don't put up with it. Don't put up with it any longer. Stand up for

yourself. Tell him, 'Do you love me? Respect me as a woman. It's not right that just because you come home drunk you beat me.' Do it for your kids.'

### Family support

In a few cases, a Cuidadora invited to the Circles a woman who said she could not attend the sessions because her husband would not allow her to. In the interviews, some participants mentioned that this might be true of other women who did not attend the Circle but did not name anyone they knew to have this issue. On the contrary, the women interviewed said they had strong family support to participate in the sessions, especially not only from their husbands but also from children and in-laws. One participant explained how her family encouraged her to go to the Circles: 'Yes, my sons and my daughter-in-law told me, 'Go to the Circle, and take some cake along so you'll bear up on the walk.' My husband also likes me to go. When I tell him I'm going to a session, he tells me to go…. He never says no, because it's better for me to be at a session than lying in my bed.' Rather than lack of their husband's permission, participants said the reasons why they missed some sessions were too much housework, illness, or heavy rain. The Cuidadoras seemed surprised by these results and had expected more women to struggle with their husbands' permission; however, it is possible that some women with this issue attended fewer than three sessions and hence were not interviewed.

### Stigmatization

Because there had been rumors that many women did not attend the Circles due to stigmatization, we specifically asked participants whether they'd heard comments from community members. Some said people were curious and asked what the Circles were for, or whether they could attend. However, interestingly, some women reported hearing negative comments not so much about mental health issues but about them going out and 'wasting time' rather than tending to their housework, kids, and husband. We also heard negative comments about how some husbands allow their wives to go out or – perhaps worse – cannot stop them from doing so. 'I was walking by and heard someone say, 'Ah, no, those women have nothing to do in their houses so they're out wasting time here.' Yes, there have been comments…. Others said, "How is it possible that the man gives so much freedom to his wife?"'

In a few interviews, participants mentioned community stigma toward mental health. 'For example, they say, "They're crazy." They say, "They're not right in the head," and other things.' In the community, the Circles were perceived as an activity organized by the health clinic, which implied that the participants had mental health problems. This assumption was not wrong as most of the women invited were the Cuidadoras' previous mental health patients. However, in general, stigma for being 'crazy' because they attended the Circles seemed minor.

Finally, according to some interviewees, the fact that only some women were invited to the Circles raised questions as to why some were invited and others not. Unlike for most community meetings where people are invited through the village loudspeaker, women were invited to the Circles privately, in their homes. 'Later this will create problems. We'll have trouble with those other women because not all women were invited to the meetings. Well, that's what people say.' So rather than fearing stigma for attending, women felt left out if they were not invited.

### No lasting friendships

Given the women's restricted social opportunities, along with research suggesting that social capital can improve mental well-being (Almedom, 2005) and provide support in intimate partner violence cases (Benavides et al., 2019), the Circles were partly intended to improve women's social networks. However, despite the positive feelings, support, and trust developed within the group, few participants met outside the Circles. One small group met regularly at religious activities before the Circles and sold food once for a Church event. Another small group visited a participant who lost a family member, a one-time event. Reasons the participants gave as to why they did not gather with other women they had not known before the Circles were that most of them were family members, some lived far away, and, in the case of two younger women, because of the age gap between themselves and the older women. So, though the women greatly appreciated the friendships they formed within the Circles (and with the Cuidadora), such friendships rarely, if ever, extended beyond the sessions.

### Discussion

#### Women want someone to talk to

Over and over, participants said that what helped them the most was having someone to talk to in confidence. They greatly appreciated the opportunity to share their experiences and to connect with women outside their family. In the Sierra Madre, it is rare for women to meet or gather in public places. While men attend *ejido*[2] meetings, practice sports, and are commonly seen in groups at the plaza or on the streets, women are expected to stay at home, with a few exceptions: taking kids to and from school, parent–teacher association meetings, church activities, birthday parties, going to the health clinic, and *ejido* work groups (e.g., cleaning the school or collecting trash). Coincidentally, women make the most of the 'official' opportunities to gather, arriving at schools early or staying late to chat with other mothers, volunteering and holding office in church groups, and seemingly even enjoying sweeping and trash collecting in the company of other women (despite being charged a fine if they fail to go).

#### Prescriptive psychoeducation versus local experience

Answers to some questions in the interview guide – indeed, the questions themselves – revealed a lack of awareness of the distinction between prescription and description. For example, when asked 'Did you learn something in the Circles about how gender affects mental health?', rather than their own experiences, the women repeated back what the Cuidadoras taught them about gender and mental health in the sessions. The women learned about women's rights, gender equality, *machismo* (sexism), gender norms, and how they 'should' be treated and respected, how they 'should' be equal partners in marriage, and how men 'should' treat women. They also learned about 'depression' and 'anxiety' and their respective symptom pools and learned to see their sadness and worries as symptoms of these labeled illnesses that can be treated with pills and therapy. They learned about self-worth, self-care, and assertiveness, concepts they were not familiar with before attending the Circles. In other words, the Cuidadoras' talks about mental

---

[2]An "ejido" is state-supported communal land with individual use, especially used for agriculture.

health and gender issues were prescriptive rather than descriptive, ideas rather than practical reality, how things should be (i.e., 'women have rights') rather than how they are. The talks included 'textbook,' widely circulated examples of gendered behavior (i.e., 'men are *machistas*') rather than the local women's perspectives and experiences. This fits with Klein and Mills (2017) description of most 'psy' therapeutic approaches, in that many of the Cuidadoras' 'lessons' focused on individual healing rather than structural change.

#### Overacceptance of psychoeducation

We found that the Cuidadoras placed more value on psychoeducation than had been intended by program staff. For instance, lessons on depression and anxiety were neither in the manual nor were many of the lessons on gender. Furthermore, there is neither explicit mention of 'rights' nor is there a recommendation for how women who experience intimate partner violence should act (the focus is on how to support other women); however, some women recalled learning from the Circles that they have the right to disobey their husbands and to stand up for themselves – which in the context of the Sierra Madre might increase women's risk of suffering violence.

The women could also have picked up such notions from CES doctors or psychologists, or from Cuidadoras during PM+ therapy. Also, like Duncan (2017), we observed that psychoeducation and information on gender-based violence circulate widely through schools, the media, the *Prospera* cash-transfer program, churches, and posters in public places (including health clinics). Still, it seems the Cuidadoras put more emphasis on prescriptive notions of mental health and gender than had been intended. And since they were not scripted in the Circles manual, they were likely acquired by the Cuidadoras during training by CES psychologists. Such training may have replicated the coloniality of the university education that psychologists receive (Capella Palacios and Jadhav, 2020; Pillay, 2017).

#### Global to local mental health

CES psychologists find themselves at the forefront of 'psychological modernization' (Duncan 2017), which popularizes globalized mental health constructs, minimizes local knowledge, and focuses on economic productivity, all in alignment with the 'scaling up' strategy of the World Health Organization that aims to increase mental health care access globally (2016; 2024). Many unintended consequences can stem from such a global health strategy, in particular from psychoeducation. Not only can psychoeducation impose 'modern' and 'foreign' views of mental health in communities but also, as affirmed by Foulkes and Andrews (2023), learning about mental health problems might cause people to internalize symptoms and bring about or exacerbate distress. Furthermore, while many global health psychological intervention manuals encourage local adaptation (e.g. WHO, 2024), the core of these interventions must still be questioned based on a deeper understanding of the local context.

The CES mental health team has an opportunity to contextualize and optimize mental health services in the local setting. After all, psychoeducation can reduce stigma (Sampogna et al., 2017) and enable people to access treatment (Henderson et al., 2017). The team constantly questions the medicalization of mental health problems, recognizes the impact of social disparities on mental health, and acknowledges that there are different 'ecologies of

suffering' (Jadhav et al, 2015) and variations in the local embodiment of suffering due to structural violence (Farmer, 1996).

The Cuidadoras, who are themselves members of the community, should continue to play an active role not only in adapting interventions but in creating them. In the pilot program, we did not observe them minimizing local knowledge as much as weaving together the 'modern' and 'local.' For instance, they encouraged conversations using local idioms of distress (Desai and Chaturvedi, 2017) and resilience (e.g. Eggerman and Panter-Brick, 2010) and constantly adapted the Circle content, changing the language, creating metaphors, and giving examples of local activities that improve mental well-being (many of them suggested by the Circle participants). These actions can also be a means of 'cultural resistance' to psychologization, as also observed by Capella (2023) in Ecuador. Finally, to avoid replicating hegemonic mental health concepts, including Western forms of psychoeducation, the mental health education and supervision they receive should encourage critical thinking.

### Women who might benefit the most do not come

For the most part, women who are currently experiencing intimate partner violence with a controlling spouse (Johnson, 2005) are not likely to attend the group. We heard from many women, including the Cuidadoras, that women with a controlling husband have difficulty leaving their house without their husband's permission. An option might be for the Cuidadora, a community health worker, or another woman to visit these women at home, strategically and discretely to avoid further violence. Furthermore, it was originally thought that the Circles might create empathy and stir participants to befriend and support women outside the Circles who needed such support, but we did not see this happen. It should be noted that while strategies for women to receive emotional and community support are crucial, adequate social protection policies to prevent gender-based violence (Cookson, et al., 2024) are needed in the Sierra Madre.

### Aftermath

As a result of the findings of this study, a revised version of Women's Circles, cocreated with the Cuidadoras, expanded to seven additional villages. As indicated by their original name, 'Embroidery Circles,' the emphasis is on the activity – an enjoyable pretext for gathering and sharing – and questions to spark conversation, rather than on psychoeducation or structured activities. While the women – including the Cuidadora – do arts and crafts, they talk (*platicar*) and listen. The Cuidadora asks open-ended questions to get and keep the conversation going. Although each session still has a theme, the facilitation guide is short and flexible, and the Cuidadora can improvise questions depending on where the conversation is headed. There are also sometimes interactive activities or relaxation exercises, and the sessions still end with refreshments. Furthermore, as many women from the first cohort wanted to continue participating, the new Circles are ongoing rather than having a set number of sessions.

### Limitations

This study was limited to a single pilot project, in four villages, that lasted 3 months. The study protocol was created and approved before the arrival of the second author. In the spirit of participatory research, the 13 interview questions were created during a group discussion with the Cuidadoras (though to mitigate the fixed nature of the questions, the interviewer, as mentioned above, encouraged

the participants to meander.) Participant responses could have been influenced by the perception that the interviewer was a member of the CES mental health team (though she said she was not). Further, the women said they felt less sad when they attended the circles, but we do not know whether it was temporary or whether overall mental health improved. Nor do we know whether women continue to use the relaxation exercises they learned in the Circles. We also do not know whether mental health improved – if it did – thanks to the Circles or to one-on-one PM+ therapy sessions with the Cuidadora, or both, or neither. For example, grief could diminish on its own, there could be a stellar coffee harvest, debts could be repaid, the health of family members could improve, etc. Finally, only about one-third of the women who were invited to the Circles attended at least three sessions; since we did not interview the others, we do not know why they did not attend except indirectly, by hearsay.

### Conclusion

The women who attended the Circles credited the sessions with improved mental well-being. However, this was likely due more to the opportunity to gather and share experiences – and have fun – than to psychoeducation and structured group activities. The Western psychological assumption that inward thinking and psychoeducation are beneficial and that symptoms of and solutions for mental illness are universal and focus on individual rather than collective care should continue to be questioned. It behoves global health organizations such as ours to reflect on the consequences of perpetuating such assumptions through the 'mental health' services we offer.

**Open peer review.** To view the open peer review materials for this article, please visit http://doi.org/10.1017/gmh.2025.15.

**Data availability statement.** The data that supports the findings of this study is not publicly available or available upon request as this information could compromise the privacy of the research participants.

**Acknowledgements.** We would like to thank the Compañeros en Salud staff members for their administrative and logistical support, the Cuidadoras for their meaningful insights into their role as observers and Circle facilitators, and their help contacting the Circle participants. We want to extend a special thank you to Fátima Rodríguez for all her mentorship and support as a program coordinator, including the IRB application. Finally, we are grateful to all the Circle participants for taking the time to be interviewed and offering us the opportunity to gather (*convivir*) with them.

**Author contribution.** A.O. designed the evaluation methodology; M.B. conducted and transcribed the interviews. Both authors did participant observation, analyzed the data, and wrote and edited the paper.

**Financial support.** This research received no specific grant from any funding agency, commercial, or not-for-profit sectors. However, Compañeros en Salud (CES) provided us with logistical and administrative support, and one of the authors receives a salary from CES. Further, CES receives unrestricted funds from its affiliate Partners in Health (located in Boston), and the CES mental health team currently receives funding from the Many Voices Foundation located in Princeton, NJ.

**Competing interest.** We declare that we have no conflicts of interest. None of the authors will receive financial gain from this publication. Although one of the authors is an employee at Compañeros en Salud, this publication will not affect her employment or salary.

**Ethics standard.** The project was approved by the Ethics Committee of the Instituto Mexicano de Trasplantes (07-18-2022, IMT-05072022-01).

Interviewed participants provided written informed consent with the signature of two witnesses.

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
