## [Reviewer Report]

This paper is an excellent example of the need for more critical scrutiny of “psychoeducation” in the mental health field around the world. The authors´ report of the “aftermath” of their study is a valuable illustration of the way critical research may help to improve specific interventions. The paper makes a relevant contribution and teach us, as other cases around the world, about the complex impact of globalized mental health interventions (both positive and negative).

METHODS

A few additional details may be included, as well as thicker descriptions of ethnographic fieldwork and interactions in the field. Did both authors conduct participant observation during the same time period? Did both live in the same village and household? Authors inform that the interview guide was designed “once the first circles were completed”; How many were the “first circles”, exactly? Regarding the analysis, the authors “deductively group excerpts by themes”; was such deductive analysis conducted only with data produced from interviews or also conducted with data from field notes? To be explicit about the pre-defined themes for analysis is important; where these the same pre-defined themes of the semi-structured interview guide? The authors declare an ethnographic approach, which is consistent with them conducting participant observation and interviews, as reported. Methods are also participatory, in the sense that “cuidadoras” took part in the design of the interview guide; and that interviewees were encouraged to talk about any topic, to dialogue, etc. I wonder how both induction (e.g. attempts to begin the study with no explicit theoretical orientation) and deduction (i.e. the themes used to analyse data) played out in this specific study, given epistemological debates around ethnographic methods (and qualitative methods, more broadly).

RESULTS:

In different sections, the authors refer to what women “said” to them. I think it would be beneficial to clarify if/when women´s words were said during semi-structured interviews; during the actual “círculos”; or in informal spaces/dialogues. Given the ethnographic approach declared by the authors, a few thicker descriptions when citing what some women said may be useful; e.g. who is the woman speaking? Her identity, maybe a protected name, what was the interaction in which her words were said/heard? What was happening? What were the authors´ impressions during such specific interactions? How such interactions may have shaped the words said by women? How did the researchers feel? I think the authors may decide to engage in such thicker descriptions or reflect on this as a potential limitation, depending on available data, the way data was gathered and analysed, or other criteria.

INTRODUCTION/DISCUSSION

Authors mention “the global authority of meta-psychoeducation”. From the context of the paper, I get a sense of what they mean by this, but they may want to consider a more explicit clarification. Due to my own positionality, I find the results to be very relevant for academic and ethical-political discussion. Specially the main finding: that healing in “círculos” came (at least fundamentally) not from globalized, structured and scaled-up “psychoeducation”, but from having space/time to gather, share and have fun together. In such context, I think that psychoeducational activities as an “excuse” to leave demanding/routine household chores is also key (as incorporated in the aftermath). Many of these conclusions are similar to what has been observed in other contexts, not only in terms of the way university training (and its coloniality) shapes professional work, but regarding experiences by working class women in diverse marginalised communities (e.g. Ecuador; Capella & Jadhav, 2020, as cited in the current version of the manuscript; Capella, 2023; India; USA; UK; México, but also many other contexts around the world). I also think that the author´s observation of “cuidadoras” weaving together the “modern” and “local” opens up further space for discussion. While the discussion, in its current form, is certainly appropriate, relevant and coherent, the authors may want to consider expanding some of the key points (e.g. by referring back to the sourced cited in their introduction; by citing a few additional sources, including updated critical discussions on “global mental health”, psychologisation, structural and cultural competence). Among other examples, authors may consider engaging with discussions on what the “social” means in the context of global mental health, and how this translates into methods for psychoeducation (e.g. Bemme & Béhague, 2024).

There are a few minor issues with style in different sections of the paper (e.g. omission of a page number in a direct quote; the size/type of fonts in a few sentences; grammar revisions needed in a few sentences, by “stigmatism” authors refer to “stigmatization”… stigma?; the interview theme “comments of others” actually refers to comments of others, or comments “by” others?).These minor issues regarding the clarity of written communication need to be revised as well.

---

## [Reviewer Report]

The paper “To gather is to heal: Women’s Mental Health Circles in rural Chiapas, Mexico” is a welcome contribution to a growing global body of scholarship of the importance of self-empowerment among communities in LMICs to deal with mental ill-health. The contribution is particularly welcome given the geographic region from which it has been generated. While the paper is well-written, it perhaps requires some additional considerations in the Discussion section. Specifically, important points are raised that could be much enriched by embedding it in existing literature. The effects of social contact and engagement on individual mental health has been well studied, with many theoretical insights produced. In particular, the unintended consequences of women attending PHC clinics for various services and finding support through interaction with other service users could be a useful addition. Comparison to similar interactions in different cultures and socioeconomic environments could add further depth.

It might be useful to reflect on the underlying reasons why “They teach us, but it doesn’t stick” emerged in terms of a programmatic intervention. There is something to say for the comparative strength of organic organisation and communal empowerment over external programmatic attempts at intervention, as well as the underlying assumptions drawn from programme designers in attempting to facilitating change. What could/should have been done differently? There seems to be a call for true participatory collaboration with participants and communities from the outset, in order to more deeply understand drivers of social forces. Further, more could be said about the prevailing discourse of psychoeducation that speaks to a tension between psyc-sciences and local knowledge.

The distinction between the “modern” and the “local” is not entirely clear.

The section “Women who might benefit the most don’t come” could be enriched by considering what has worked in other contexts, highlighting how women in such vulnerable positions can be included and empowered.

---

## [Reviewer Report]

The authors have adequately addressed the points raised in the previous review. I think that this study makes an important contribution and that it should be published. That being said, I still miss some more detail in the reporting of the analysis, as well as more situated and relatively thicker descriptions. Likewise, I insist that the discussion could open up more links between the case studied by the authors, and the most up-to-date debates on psychoeducation and the strengths/limitations/iterations of “global mental health.” I also understand that there are space limitations, and that the authors are free to decide which points to discuss or not.They could consider further refining some of these points, or, at the very least, including a brief paragraph (or even sentence) that, reflexively, makes explicit the “limitations” of the study (methodological, reporting of results, and/or discussion, etc.). I hope this feedback helps the authors to make minor changes to what already is an interesting and relevant paper.

---

## [Reviewer Report]

The authors have sufficiently addressed the points I have raised in previous reviews. This is a valuable and relevant study, with strengths and limitations like any study. I appreciate the opportunity to review it, as I was able to learn more about the phenomenon under study. The article contributes to more complex and situated understandings of “psychoeducation” and “mental health,” considering specific contexts of analysis and intervention. I recommend that it be accepted for publication.